# Perivascular Adipose Tissue and Perivascular Adipose Tissue-Derived Extracellular Vesicles: New Insights in Vascular Disease

**DOI:** 10.3390/cells13161309

**Published:** 2024-08-06

**Authors:** Smara Sigdel, Gideon Udoh, Rakan Albalawy, Jinju Wang

**Affiliations:** 1Department of Biomedical Sciences, Joan C Edwards School of Medicine, Marshall University, Huntington, WV 25755, USA; sigdels@marshall.edu (S.S.); udoh3@marshall.edu (G.U.); 2Department of Internal Medicine, Joan C Edwards School of Medicine, Marshall University, Huntington, WV 25755, USA; albalawy@marshall.edu

**Keywords:** perivascular adipose tissue, extracellular vesicles, vascular diseases, obesity, diabetes

## Abstract

Perivascular adipose tissue (PVAT) is a special deposit of fat tissue surrounding the vasculature. Previous studies suggest that PVAT modulates the vasculature function in physiological conditions and is implicated in the pathogenesis of vascular diseases. Understanding how PVAT influences vasculature function and vascular disease progression is important. Extracellular vesicles (EVs) are novel mediators of intercellular communication. EVs encapsulate molecular cargo such as proteins, lipids, and nucleic acids. EVs can influence cellular functions by transferring the carried bioactive molecules. Emerging evidence indicates that PVAT-derived EVs play an important role in vascular functions under health and disease conditions. This review will focus on the roles of PVAT and PVAT-EVs in obesity, diabetic, and metabolic syndrome-related vascular diseases, offering novel insights into therapeutic targets for vascular diseases.

## 1. Introduction

Perivascular adipose tissue (PVAT) is an adipose tissue deposit surrounding most of the vasculature. PVAT releases a variety of adipocytokines and chemokines [1,2,3,4]. Under physiological conditions, PVAT regulates vascular tone and reactivity by releasing various biologically active factors, adipocyte-derived relaxing factors, and perivascular-derived relaxing factors, which have anticontractile properties. However, in pathological conditions, such as obesity, diabetes, and metabolic syndrome, PVAT becomes dysfunctional. The secretory profile of PVAT changes with a reduced secretion of vaso-relaxing factors, increased vasoconstricting factors, increased immune cell infiltration, and vascular smooth muscle cell proliferation. These changes lead to PVAT inflammation, thereby detrimentally affecting vasculature function. Studies found that PVAT mass is increased in obesity in both animal models and humans [5,6,7]. In obese patients, PVAT mass correlates with visceral adipose tissue mass, a known predictor of the development of metabolic disease [6,8]. PVAT volume alone is associated with hypertension and aortic and coronary calcification [6,9]. All of these findings suggest the involvement of PVAT in vascular physiology and pathology.

Inter-cell signaling between PVAT and the vasculature remains a mystery. Recent studies have suggested that, apart from the secretory cytokines and chemokines factors, PVAT also secretes extracellular vesicles (EVs) [10,11]. EVs are cellular-derived nano-sized particles, carrying biologically active molecules that can regulate cell function. EVs can be transferred horizontally to adjacent cells (paracrine signaling) or distant cells and organs (endocrine–inter-organ signaling). They are key players in various physiological and pathological processes, ranging from cancer metastasis to cardiovascular diseases [12]. In this study, we reviewed the existing literature regarding the roles of PVAT and their derived EVs in the pathogenesis of vascular diseases. We discussed the therapeutic potential of targeting PVAT and PVAT-EVs in vascular diseases and the challenges in this field.

## 2. PVAT: Anatomy and Physiology

PVAT is a specialized type of fat tissue that adjacently surrounds blood vessels such as the aorta, coronary arteries, small and resistance vessels, and vasculature of the musculoskeletal system [13], but is absent among cerebral vessels [14]. Though PVAT is universal in that it surrounds vasculature, its volume and composition vary widely depending on physiological and pathological conditions. This demonstrates how versatile and diverse PVAT may be, and what innumerable effects it may elicit on surrounding tissue. Like most adipose tissue, PVAT could be white adipose tissue (WAT) or brown adipose tissue (BAT). For instance, the thoracic aorta is surrounded by brown PVAT while the abdominal aorta PVAT shows the WAT-like phenotype [15]. Usually, WAT performs as an energy reserve and is required for maintaining human homeostasis. WAT is regarded as the traditional “unhealthy” fat, as experimental and clinical studies have shown that it is positively related to cardiovascular diseases [16]. WAT has unilocular lipid droplets, fewer mitochondria, and higher lipid storage capacity [17]. The unique surface markers of white adipocytes include resistin, ASC-1, and FAB4 [18]. Different types of WAT include visceral adipose tissue and subcutaneous WAT. Visceral WAT has been shown to negatively impact health due to its inflammatory characteristics, which are linked to insulin resistance and cardiovascular events. On the other hand, subcutaneous WAT has a higher expression of uncoupling protein 1 (UCP-1) which reflects its ability to undergo browning [19]. Unlike WAT, BAT is considered “good” fat and is utilized for heat production and anti-pathogenic effects. BAT contains multilocular lipid droplets, a higher number of mitochondria, and thermogenic capacity due to elevated UCP1 amounts anchored in its mitochondrial inner membrane [20]. Recently, it has been made apparent that the dichotomy between WAT and BAT is not so stark. Studies show that WAT is manageable in conditions such as cold or adrenergic stimulation [21] and is capable of “beiging” or adopting traits typically characteristic of BAT such as expressing the UCP1 associated with thermogenic activity [22,23].

The main cells that compose adipose tissue are adipocytes. Besides adipocytes, preadipocytes, fibroblasts, capillary endothelial cells, macrophages, and stem cells are also present in adipose tissue. Similarly, PVAT also contains adipocytes, pre-adipocytes, immune cells, fibroblasts, endothelial cells, stem cells, etc. The variety in the cellular composition of PVAT corresponds to the diverse biomolecules it releases. PVAT can release vasodilator and anticontractile factors including adiponectin [24] and angiotensin 1–7 [25]. PVAT can also release anti-inflammatory factors (angiopoietin-like protein 2, IL-10), pro-inflammatory factors (IL-6 and TNF-a), microRNAs [26], reactive oxygen species, and metabolites [27]. It is believed that the functions of PVAT are closely related to its released factors via either paracrine or endocrine pathways. Such endocrine and paracrine effects correspond with diverse effects on vasculature [27,28]. For example, adiponectin levels are inversely proportional to visceral fat accumulation, insulin sensitivity, and risk of developing type 2 diabetes (T2D) [29]. In obese individuals, especially those with central adiposity with a higher waist-to-hip ratio, the overall adiponectin level is less when compared with non-obese individuals [30]. André Tchernof and colleagues have reported a negative correlation between adiponectin release and omental adipose tissue area/total body fat mass/omental adipocyte diameter in women undergoing elective surgery. This correlation was not present in the subcutaneous adipose tissue [31]. The possible explanation is that smaller, more insulin-sensitive omental adipocytes are more efficient in releasing adiponectin when compared with larger omental fat cells. In addition, adiponectin also plays important roles in atherosclerotic plaque formation [32], fatty acid oxidation, and glucose uptake in muscle tissue [33].

Besides regulating the vasculature dilation and contraction, PVAT has also been shown to be implicated in neointimal formation, vascular calcification, and arterial stiffness as well as the pathogenesis of aneurysm [34], although the detailed mechanisms are not fully understood. Additionally, PVAT affects muscle functions. Turaihi and colleagues found that removing PVAT from muscle blood vessels decreased glucose uptake by muscles, upregulated proteins associated with cell stress and inflammation, and downregulated essential mitochondrial proteins related to glucose metabolism [35], suggesting the roles of PVAT in the vasoreactivity in muscles. Notably, studies have found that the crosstalk between the PVAT and typical cells belonging to the vasculature is not one-sided. In times of oxidative stress and inflammation, the vascular walls may release bioactive molecules, such as lipid peroxidation products and inflammatory cytokines, which stimulate the anti-oxidative and anti-inflammatory functions of PVAT. For instance, Komuro et al. have shown that beiging of PVAT fine-tunes inflammatory response in a mouse model of endovascular injury [36]. Our recent study has revealed that exercise intervention can alleviate inflammation and oxidative stress in thoracic aorta PVAT in type 2 diabetic mice [37]. It is worthwhile to note that gender can influence PVAT functions. Clinical studies have found that perivascular and pericardial adipose tissue are increased in women after menopause, and that the volume of aortic PVAT positively correlates with the reduction in estradiol [38,39]. In addition to the abovementioned functions of PVAT, various types of cells in PVAT can release EVs. This will be the main point of discussion in this review. 

## 3. Extracellular Vesicles (EVs)

EVs are small, membrane-bound vesicles released by cells into the extracellular environment. There are three main classifications of EVs: exosomes, microvesicles (or microparticles), and apoptotic bodies. In general, EVs, including PVAT-derived EVs (PVAT-EVs) have unique variations in properties like biogenesis, size, content, and function, which vary on their subtype, cellular origin, and cellular status. Previous studies have suggested adipose tissue-derived EVs are intercellular communicators for systemic metabolism regulation. They can also influence insulin sensitivity, inflammation, and energy homeostasis [40]. Recently, a growing body of evidence has begun to suggest that PVAT-EVs play important roles in regulating vascular homeostasis, blood endothelial and vessel functions, influence vascular tone, and metabolic inflammation, as well as being involved in the development of vascular diseases such as atherosclerosis [41,42]. 

### 3.1. PVAT-Derived Exosomes and Microvesicles: Biogenesis and Size 

The biogenesis and release of PVAT-EVs share biological mechanisms similar to the general cellular EV release [43]. Exosomes are EVs typically ranging from 30 to 150 nm in diameter. The exact mechanism of exosome biogenesis from PVATs or other tissues or cells is incompletely understood. So far, most studies have shown that exosomes are formed through the endosomal membrane’s inward budding, which produces intraluminal vesicles (ILVs) within multivesicular bodies (MVBs) [44,45,46]. The MVBs subsequently fuse with the plasma membrane, releasing the intraluminal vesicles as exosomes. The endosomal sorting complex required for transport (ESCRT), which includes four protein complexes (ESCRT-0, -I, -II, and -III) and accessory proteins such as ALIX and VSP4, is one of the major biogenesis mechanisms of exosomes [47,48]. It is reported that this complex causes ILVs to be enclosed within MVBs by recognizing their ubiquitinated membrane proteins and promoting their entry into the MVBs. Studies have also proposed that ILVs may be formed through alternate mechanisms involving tetraspanin (e.g., CD9, CD63, CD81) and lipid rafts [49]. 

Microvesicles, called microparticles or ectosomes, are EVs typically ranging from 100 to 1000 nm in diameter [12]. The outward budding and fission of the cellular plasma membrane form microvesicles [50,51,52,53]. Studies suggest that shifts in intracellular calcium levels due to factors such as mechanical stress, inflammation, or apoptosis are one of the causes of this process. These shifts are thought to activate enzymes like calpain, responsible for cytoskeletal rearrangement, cytoskeletal protein cleavage, and membrane detachment/blebbing [50,51,52,53]. It is reported that, during membrane detachment/blebbing, enzymes like scramblases and lipases promote phospholipids, such as phosphatidylserine, on the outer leaflet of the budding membrane, inducing membrane curving and formation of vesicles. These budding vesicles are thought to take up biomolecules from the cytoplasm before being split and released from the cell membrane through the contraction of actin and myosin filaments. 

### 3.2. PVAT-Derived Exosomes and Microvesicles: Uptake Pathways and Functions

The mechanisms of EV uptake vary depending on the specific type of EV and the recipient cell. Studies have demonstrated that multiple pathways, such as direct interaction, the fusion with the plasma membrane that is mediated by SNAREs and Rab proteins [54], cell membrane fusion [55], receptor-mediated signaling, and endocytosis, are responsible for the exosomes and microvesicles that alter the functions of recipient cells in the target tissue. For instance, during cell–cell interaction, exosomes and microvesicles fuse with target cell membranes and directly deliver their contents into the cytoplasm. During receptor-mediated signaling, EVs utilize their surface proteins to communicate with receptors on recipient cells. The endocytosis method involves exosomes and microvesicles becoming engulfed by the recipient cell membrane and brought into the cell. The transmembrane ligands on the exosome surface can bind directly with the surface receptors on recipient cells and activate downstream signaling cascades to activate target cells. To exemplify, Guan et al. have found that umbilical cord blood-derived exosomes expressing MHC-I and MHC-II stimulate T cell proliferation to produce anti-tumor activity [56]. Mesenchymal stem cells can uptake PC12-derived exosomes via clathrin-mediated endocytosis and micropinocytosis pathways [57]. Bone marrow stromal cells could uptake myeloma-derived exosomes using a caveolin-dependent pathway, associating micropinocytosis, and membrane fusion pathway [58]. All of these findings suggest that the internalization pathways for exosomes and microvesicles vary, depending on the EV subtype and target cells. Further, the EV cellular origin and cellular status also significantly affect up-taken EV and its effects. For instance, cardiac progenitor cell-secreted EVs are preferentially taken up by cardiomyocytes and endothelial cells when compared with EVs that are released by fibroblasts and bone marrow stem cells [59]. Papini et al. found that exosomes from sulforaphane pre-treated fibroblasts (an edible class I HDAC inhibitor) exhibited approximately three times more selective uptake by cardiomyocytes when compared with exosomes from untreated cells [60]. It is worthwhile to note that epigenetic food compounds such as sulforaphane may influence the acetylome of fibroblasts and cardiac progenitor cells [61], which in turn may modulate EV biogenesis and release from PVAT fibroblasts. These findings inspire investigation into PVAT-EV studies, as adipocyte progenitor cells and fibroblasts are present in PVAT [62]. Understanding whether there is a specific incorporation preference for adipose progenitor cell-derived EVs or fibroblast-derived EVs will help us manage PVAT-induced vascular diseases.

As a result of the ability to carry and transport biomolecules [12], exosomes and microvesicles have been found to participate in various biological processes. Besides removing cellular detritus, the major role of exosomes and microvesicles is the conducting of intercellular communication [63,64]. Exosomes and microvesicles, including PVAT-EVs, have been shown to deliver proteins, lipids, and nucleic acids to recipient cells, which affects target cell behavior and leads to various cascading effects. For instance, large adipocyte-derived EVs can transfer lipogenic information to stimulate lipid storage in small adipocytes [65]. PVAT-EVs from high-fat diet mice could trigger early-stage vascular remodeling in C57BL/6J mice, mainly ascribed to their carried microRNA-221–3p [66]. Adipose tissue-derived EVs from ob/ob mice have been shown to induce insulin resistance and promote pro-inflammatory cytokine secretion in control mice [10]. Zhang et al. have found that adipocyte-derived EVs from obese mice carrying microRNA-155 induced macrophage-to-pro-inflammatory phenotype change [67]. Given their role in intercellular communication, exosomes and microvesicles are shown to be implicated in various physiological and pathological processes within the body [12,63,64]. These processes include immune regulation, tumor progression, neurodegenerative disease propagation, coagulation, and tissue regeneration/repair [12,68,69]. Regarding immune regulation, research has revealed that exosomes and microvesicles can influence immunoregulation via antigen presentation and the release of immunoregulatory proteins such as cytokines [70,71]. These proteins modulate the immune activity by stimulating or suppressing it. In the context of tumor progression, studies suggest that, when exosomes and microvesicles are released by cancer cells, they convey oncogenic proteins and miRNAs that can alter and recruit local cells or cells in other regions of the body [72,73]. This enables cancer cells to create a metastasis microenvironment, which can cause effects like cell proliferation, angiogenesis, and immune system evasion. Research also reveals that cancer-derived EVs can induce immunosuppressive responses that promote tumor progression. As a result of their participation in cancer progression, exosomes and microvesicles can function as biomarkers for diagnosing and monitoring cancer or serve as therapeutic targets for cancer. Several groups have reported that exosomes and microvesicles can induce immune responses against tumors via antigen presentation and could be used as drug-delivery vehicles. In neurodegenerative disease, exosomes can propagate misfolded proteins such as amyloid-beta, tau, and alpha-synuclein [74]. These proteins promote the progression of diseases like Alzheimer’s, Parkinson’s, and Huntington’s. Exosomes have also been found to deliver inflammatory molecules such as cytokines, which induce a neuroinflammatory response when presented to neuroimmune cells like microglia. Beyond promoting neurodegenerative disease progression, exosomes have been revealed to have the potential to provide neuroprotection against neurodegenerative diseases. Research indicates that exosomes from mesenchymal stem cells can carry neuroprotective biomolecules such as microRNAs, that promote neuronal survival and repair in ischemic rats [74]. Our group has previously demonstrated that endothelial progenitor cell-derived microvesicles can elicit favorable effects in a diabetic ischemic mouse model [75]. More recently, we have found that exercise intervention can modulate the level and functions of endothelial progenitor cell-derived EVs in hypertensive conditions [76], indicating a promising therapeutic target for hypertensive-related cerebrovascular diseases such as ischemic stroke.

Microvesicles have been found to play a role in blood coagulation. Several groups have demonstrated that microvesicles from platelets carry pro-coagulant phospholipids and phosphatidylserine and tissue factor [68,69]. These molecules influence the coagulation process by promoting thrombin production and clot formation, which is important for coagulation processes such as hemostasis and thrombosis. Research also indicates that EVs may play a crucial part in tissue regeneration and repair due to their ability to carry and transport biomolecules associated with this process, such as growth factors, cytokines, and microRNAs [77]. Studies have found that EVs rich in these molecules, such as those from mesenchymal stem cells and endothelial progenitor cells, can promote regeneration and repair in damaged tissue by inducing activities, cell growth/proliferation, and cell survival [78,79,80]. They have also been found to aid regeneration and repair by regulating immune responses [77]. Increasingly, studies have indicated the effects of these EVs in improving various injuries such as wounds, heart, and liver injuries [77]. For example, exosomes from mesenchymal stem cells appear to improve cardiovascular health in rats after myocardial infarction by stimulating angiogenesis and inhibiting apoptosis [81]. Similarly, endothelial cell-derived microparticles could contribute to wound healing and tissue regeneration by promoting angiogenesis [82].

Overall, exosomes and microvesicles are EVs formed via the inward and outward budding of the plasma membrane, respectively. They are involved in various biological processes, making them important research subjects, as gaining a deeper understanding of their roles would improve their capacity for use in diagnosis and therapy.

## 4. Apoptotic Bodies: Biogenesis, Size, and Function

Apoptotic bodies are vesicles ranging from 500 to 2000 nm in diameter [83], and are the largest EVs in size. They form during apoptosis. The formation process begins with the initiation of apoptosis, triggered by internal stimuli (such as DNA damage) or external stimuli (such as cytokines) [84,85]. During this event, caspases, enzymes that induce the disassembly of cellular components, cause effects such as cell shrinkage, fragmentation, and membrane blebbing. As apoptosis progresses, these blebs split into membrane-bound apoptotic bodies that contain sections of the cytoplasm. Apoptotic bodies contain many cellular components, such as organelles, DNA fragments, protein fragments, and histones [12]. Research indicates that the primary role of apoptotic bodies alongside intercellular communication is preventing the release of harmful cellular contents. During apoptotic cell clearance, apoptotic bodies containing cellular materials could be ingested by phagocytes [84]. These play a major role in maintaining tissue homeostasis by safely sequestering and removing cellular debris. They are involved in various processes, such as apoptotic cell clearance, immunoregulation, and tissue remodeling [84,86,87]. Apoptotic bodies have also been found to carry immunoregulatory materials that can influence anti-inflammatory responses, immune tolerance, and immune homeostasis [87]. These findings illustrate their potential biological functions.

In summary, exosomes, microvesicles, and apoptotic bodies are PVAT-EVs with different properties such as biogenesis, size, content, and function (Figure 1). They exist in biological fluids such as blood and urine. The components and functions of EVs are distinct for each EV cell source and cellular status. Upon secretion, PVAT-EVs could mediate both paracrine and endocrine signaling by conveying their cargo, which consequently regulates the behavior of recipient cells [88]. Therefore, they have been shown to play crucial roles in a wide range of physiological and pathological processes, such as intercellular communication, immune modulation, apoptotic cell clearance, and vascular diseases, a function which has been largely ascribed to their various biomolecule cargo, including proteins, non-coding microRNAs, lipids, etc., making them important subjects for research. 

## 5. PVAT and PVAT-EVs in Obesity and Diabetic-Related Vascular Dysfunction

Obesity is characterized by excessive fat accumulation in adipose tissues leading to weight gain. It is a major risk factor for T2D and contributes to its severity. Diabetes mellitus is a comprehensive term for insulin-related metabolic diseases. Diabetes mellitus affects approximately ten percent of Americans [89] and about 500 million people worldwide. By 2030, this number is estimated to reach 578 million. By 2045, this proportion will reach 700 million [90]. Type 1 diabetes (T1D) is an autoimmune disorder in which beta-cell-induced destruction of insulin incites an insufficiency of the protein. It is typically unpreventable and diagnosed in adolescence [91]. T2D, in contrast, is preventable and often presents as selective insulin resistance resulting from poor diet and exercise choices. Diabetes and its comorbidities, such as obesity and dyslipidemia, induce endothelial dysfunction, inflammation, and blood coagulation. They are implicated in cardiovascular diseases (CVD) like myocardial infarction and cardiomyopathy [92]. In fact, for individuals with T1D or T2D, CVD is the primary cause of death [93]. Those with T1D are more prone to coronary artery disease and peripheral artery disease, while those with T2D are more likely to suffer from atherosclerosis and stroke [94]. 

Increasing evidence suggests that PVAT undergoes morphological changes in response to obesity and diabetes. A study implementing data from the Framingham Heart Study found that thoracic peri-aortic PVAT is correlated with body mass index and CVD [6]. Wang and colleagues found that the mesenteric PVAT mass increased following 10 weeks of diet-induced obesity in male C57BL6/J mice [95]. Similarly, Juha et al. observed that 8 months of a high-fat diet can increase abdominal aorta PVAT by three-fold in C57BL/6 mice [5]. On a more specific level, white PVAT around the abdominal aorta and iliac arteries increase by 1.9 and 1.7-fold respectively, following a 7-week high-fat diet on Wistar rats [96]. In a monogenic diabetic G protein-coupled estrogen receptor (GPER)-deficient mouse model, 12-month-old mice demonstrated a 3.6-fold increase of PVAT compared with wild-type control [97]. This study also identified PVAT as a novel regulator of arterial vasoconstriction, which acts through the release of adipose-derived contracting factors and likely contributes to increased vascular tone by antagonizing vasodilation. This is also mirrored in the hypertrophy of the adipocytes of abdominal PVAT in obesity conditions [98]. Increased PVAT mass around vasculature is associated with decreased insulin sensitivity [99]. 

Besides mass changes, PVAT undergoes functional alterations in obesity and diabetic conditions. For instance, obesity can also promote the phenotypic shift of thoracic aorta PVAT from brown to white adipocytes [100]. PVAT in aged, high-fat diet conditions demonstrates increased type 1 macrophage infiltration [101]. Such PVAT dysfunction is a major contributing factor to obesity and diabetic-associated vascular dysfunction. Several groups have observed that a high-fat diet can induce changes in PVAT functions. Bussey et al. have found that the dilation capability of PVAT, acting contrary to administered norepinephrine, is lacking in obese, high-fat diet conditions, and weight loss can reduce the inflammatory conditions of PVAT during obesity in rats [102]. Meanwhile, they found that the inflammatory factor expression was altered in obesity conditions by upregulated TNF-alpha and downregulated eNOS, the latter of which holds a protective function against oxidative stress. Chatterjee and colleagues have reported that a two-week high-fat diet can induce significant downregulations of adiponectin, PPAR-gamma, and FABP4 while upregulating leptin and MIP1-alpha and promoting inflammation in wild-type mice [103]. Almabrouk et al. found that a high-fat diet can inhibit the anticontractile activity of aortic PVAT [104]. Similarly, another study has found that nitric oxide released by PVAT, which in healthy conditions contributes to dilation of the vasculature, is remarkably decreased in obese conditions in a rat obesity model [105]. Intriguingly, the offspring of high-fat diet female mice also demonstrate decreased PVAT anticontractile activity [106]. In T2D conditions, PVAT also exhibits macrophage infiltration and changes to a vasoconstriction phenotype that aggregates endothelial dysfunction [107]. PVAT secretes less adiponectin which directly or indirectly contributes to inflammation, oxidative stress, and insulin resistance. Meijer et al. have revealed that PVAT from obese mice inhibits insulin-induced vasodilation, which could be blocked by targeting the inflammatory JNK pathway [108]. Our group has demonstrated that thoracic PVAT from diabetic mice has lower levels of adiponectin and IL-10, with increased expressions of oxidative stress, IFN-r, TNF-⍺, and IL-6 inflammatory cytokines, which influence aorta function in T2D diabetic mice [37]. 

The effect of clustered cardiovascular risk factors on PVAT function has also been studied. DeVallance et al. have reported that metabolic syndrome impacts thoracic aorta PVAT function, as evidenced by increased ROS production and pro-inflammatory cytokines, associated with decreased endothelial-dependent dilation function in aorta in obese Zucker rats [109]. Osaki et al. have found that endothelial-dependent relaxation is compromised in a high-fat and high-sucrose diet-induced metabolic syndrome mouse model. The impaired endothelial-dependent relaxation is caused by increased superoxide production from the aorta and PVAT [110]. Similarly, PVAT-derived oxidative stress and inflammation have been implicated in vascular dysfunction in a rat model of metabolic syndrome as reflected by increased ROS production, pro-inflammatory changes, and decreased endothelial-dependent dilation in vasculature [7]. Notably, the altered function of PVAT may be a link between obesity and hypertension, but the detailed mechanisms are incompletely understood. Marchesi and colleagues have reported that the immune cells in PVAT, such as macrophages/monocytes, can contribute to hypertension by administering oxidative stress and related inflammation. They showed that PVAT loses its ability to influence vasodilation in New Zealand obese mice that are predisposed to obesity, hyperglycemia, hyperinsulinemia, and hypertension [7]. Takemoria et al., have demonstrated that the removal of PVAT corresponds with increased blood pressure, alluding to the critical role that PVAT plays in blood pressure regulation [111]. Table 1 summarizes the potential changes of PVAT in obesity, T2D, and metabolic syndrome conditions.

Exercise is a non-pharmacological intervention approach to manage metabolic dysfunctions such as obesity and T2D. Exercise has been shown to restore TNF-⍺ levels, elicit anti-contractile effects and regulate adiponectin release [112]. The influences of exercise intervention in PVAT are multi-faceted. Beyond promoting adiponectin production by PVAT [113], exercise intervention can attenuate immune cell infiltration into PVAT, thereby improving vascular function [114]. Saxton and colleagues have found that exercise reduces PVAT inflammation and increases β3-adrenoceptor and OCT3 expressions to improve PVAT function in obesity [112]. Our group has reported that treadmill exercise can modulate macrophage phenotype, decrease superoxide production in PVAT, and increase NO levels in the aorta in T2D mice [37]. Similar findings show that exercise intervention can increase UCP-1 expression in the mesenteric artery PVAT in rats [115] and increase eNOS expression and phosphorylation in obese rats [113]. Additionally, myokines, such as FGF21 and irisin, released from skeletal muscle can regulate PVAT function via paracrine [114]. These findings suggest the potential of targeting PVAT to treat vascular complications of obesity and diabetes.

Recently, increasing evidence indicates that adipocyte-derived EVs link obesity and diabetes, and their comorbidities. EVs released from adipose tissue-derived cells in obesity or diabetic conditions, including adipocytes and macrophages, play a role in multiple manners, including regulating inflammation, insulin sensitivity, and vasculature functions. Wang et al. have reported that EVs and their cargo microRNAs mediate adipose tissue and brain inter-organ communication, inducing the synaptic damage and cognitive impairment associated with insulin resistance, and thus providing a promising strategy for pharmaceutical interventions for cognitive impairment in diabetes [116]. Ying and colleagues have revealed that obese adipose tissue macrophage-derived EVs containing microRNA-155 can modulate insulin sensitivity [40]. Furthermore, Deng et al. have revealed that intravenous administration of adipocyte-EV-induced systemic insulin resistance in lean mice, leading to altered insulin signaling in visceral adipose tissue, liver, and muscle, and enhancing the recipient animals’ inflammatory profile [10]. Camino and colleagues have reported that the EVs that are shed by pathological adipocytes could spread pathology by stimulating IL-6 and TNFα expression and promoting macrophage inflammation in recipient healthy adipocytes [117]. Circulating adipose-derived EVs have been shown to transport microRNA-99b, which is involved in regulating liver fibroblast growth factor 21, contributing to the control of metabolic homeostasis and systemic insulin resistance [26]. EVs from obese subcutaneous or obese visceral deposits contain reduced levels of vascular endothelial growth factor and matrix metalloproteinase-2, suggesting that they may have lowered pro-angiogenic potential [118]. However, EVs released from adipose tissue-derived stem cells are rich in microRNAs and can promote endothelial cells’ migration and invasion abilities [119]. Hartwig et al. have identified 897 adipokines in adipocyte-derived EVs isolated from lean or overweight women. These adipokines have been found to be strongly associated with human metabolic diseases, including metabolic diseases like diabetes mellitus, glucose metabolism disorders, and metabolic syndrome [120]. Another group has reported that subcutaneous adipocyte-derived EVs of obese patients are enriched in proteins implicated in fatty acid oxidation [121]. 

Li and colleagues have found that PVAT-EVs from high-fat diet mice carrying microRNA-221–3p could trigger early-stage vascular remodeling in C57BL/6J mice, suggesting the roles of PVAT-EVs in response to obesity-associated inflammation and vascular remodeling [66]. Deng et al. have demonstrated that ob/ob mouse adipose tissue-derived EVs could induce insulin resistance, promote differentiation of bone-marrow-derived monocytes into macrophage, and increase pro-inflammatory cytokine secretion in control mice [10]. Large adipocyte-released EVs can stimulate lipid storage in small adipocytes by mediating the horizontal transfer of lipogenic information to promote lipogenesis and adipocyte hypertrophy [65]. Zhang et al. have found that adipocyte-derived EVs from obese mice induce M1 macrophage phenotype through secreted microRNA-155 [67]. It has been observed that EVs secreted by obese adipose tissue stem cells contain impaired levels of microRNA-126 [122]. As exercise can restore the vasodilation functions of PVAT [37,112], the question of whether exercise could regulate the release and cargo of PVAT-EVs requires further investigation. Additionally, as discussed above, Chang and colleagues have revealed that the functions of PVATs vary from location to location, which is different from classical adipose tissue. Whether exercise intervention affects PVAT differently in different locations requires further investigation. 

Overall, our knowledge of PVAT and PVAT-EVs has grown concurrently with the increasing prevalence of obesity and diabetes. Current findings (Table 2) suggest that obesity and diabetes could impact the functions of PVAT and the cargo package of PVAT-EVs. Due to the proximity of PVAT to the associated blood vessels, PVAT and PVAT-EVs could be a novel therapeutic target for interference with obesity and diabetes-related vascular diseases. 

## 6. PVAT and PVAT-EVs in the Pathogenesis of Atherosclerosis

Atherosclerosis, a progressive cardiovascular disease, is characterized by the development of lipid-comprised and fibrous lesions and calcification of the internal walls of the vasculature. The pathogenesis of atherosclerosis is rooted in inflammation. Major risk factors of atherosclerosis include increased blood cholesterol by way of low-density lipoprotein (LDL) collection in vessels, hypertension, illicit substance misuse, and metabolic disorders such as diabetes mellitus [123]. The pathogenesis of atherosclerosis begins with stress applied to the endothelium, which serves as the barrier between vessel interiors and deeper layers, including vascular smooth muscle cells (VSMCs), adventitia, and basement membranes, which further separate vasculature from the interstitial fluid and tissue [124]. When stress is applied to endothelial cells, they can change from flat, long shapes into round shapes, which makes the walls more likely to be constricted, undergo stress, and be prone to lipid uptake and immune cell adhesion [125]. M1 macrophages can upregulate the movement of LDLs into the intima layer, and the cholesterol transported within LDLs promotes M1 macrophages to adopt the foam-cell phenotype. VSMCs can also uptake LDLs to adopt foam cell phenotypes, primarily in the coronary artery. As atherosclerosis progresses, this fatty area becomes known as a necrotic core, which becomes capped by fibers [125]. Continuing disease progression is characterized by the increased apoptosis of macrophages and VSMCs due to oxidative stress and lack of nutrients. Consequently, inflammation prevails, eventually leading to calcification of the necrotic core. If the area ruptures, thrombosis may develop, causing ischemia and damage [125].

Increasingly, studies have demonstrated that PVAT is a double-edged sword. Under physiological conditions, PVAT mainly secretes anti-inflammatory adipokines mediators such as adiponectin, omentin, fibroblast growth factor-21, nitric oxide (NO), PVAT-derived relaxing factors, and IL-10 [126]. Under disease conditions, PVAT secretes a large amount of inflammatory adipokines mediators, such as leptin, tumor necrosis factor-α (TNF-α), monocyte chemoattractant protein-1, interleukin-6 (IL-6), interleukin-1β (IL-1β), etc. [127]. PVAT could elicit favorable influences, such as anti-atherogenic effects [128], stemming from its anti-contractile and anti-inflammatory effects. The selective deletion of PVAT in mice can impair endothelial function and augment atherosclerosis [66], suggesting a protective role of healthy PVAT. On the other hand, in disease conditions like diabetes or obesity, PVAT becomes inflamed and dysfunctional and is implicated in the pathogenesis of atherosclerosis and vascular remodeling [129]. Horimatsu and colleagues have demonstrated that the transplantation of white PVAT isolated from high-fat-fed wild-type mice onto the abdominal aorta of LDL-receptor knock-out mice promotes endothelial dysfunction and atherosclerosis in the remote thoracic aorta [130], suggesting the remote effects of PVAT on the vessel. This study also supports the concept that PVAT does not require direct contact to influence the vasculature, pointing to the potential role of PVAT-EVs. 

Accumulating evidence suggests the importance of adipose tissue-derived EVs in preserving cardiovascular physiology. Zhao and colleagues have reported that a period of exercise of 4 weeks can induce the enlargement of brown adipose tissue and alter the microRNA profiles of the adipose tissue released EVs. Furthermore, they have demonstrated that the adipose-tissue-released EVs could regulate cardiomyocyte survival and exhibit exercise cardioprotection in the context of myocardial ischemia/reperfusion injury [131]. Similarly, another study has shown that brown adipose tissue-derived EVs mediate the communication from brown adipose tissue to cardiac myocytes and cardiac fibroblasts. It has been shown that treatment with exosomes from brown adipocytes treated with a β3-AR agonist conferred protection against angiotensin II-induced cardiac remodeling. The authors have further pinpointed iNOS as a critical cargo component of EVs derived from beige adipocytes with β3-AR knockdown, and that it contributes to cardiac fibroblast dysfunction and cardiac remodeling [132]. Additionally, medications such as Ticagrelor, which reversibly targets ADP-mediated G protein-coupled purinergic receptor P_2_Y_12_, and has been widely used in patients with acute coronary syndrome and myocardial infarction, could modulate the functions of cardiomyocyte-derived EVs on hyperglycemic cardiomyocytes through the alleviation of oxidative and endoplasmic reticulum stress [133]. However, whether it could affect the texture and cellular proliferation of PVAT and influence the release and function of PVAT-EVs requires more investigation. 

Beyond the intercellular communication with cardiomyocytes, it is speculated that the effects elicited by PVAT might be attributed to the crosstalk between PVAT-EVs and cells residing within the vasculature. Earlier studies have demonstrated that EVs travel between adipose tissue and endothelial cells. Flaherty III et al. have found that adipocyte EVs are both an alternative pathway of local lipid release and a mechanism by which parenchymal cells can modulate tissue macrophage differentiation and function [134]. Adipose-derived mesenchymal stem cell-derived exosomes could protect endothelial cells from atherosclerosis, an ability that is ascribed to their carried microRNA-342–5p [135]. Exosomes from adipose-derived stem cells can promote VEGF-C-dependent lymphangiogenesis by directly downregulating Smad7 and regulating TGF-b/Smad signaling in lymphatic endothelial cells [136]. Indeed, several studies have revealed that cardiac lymphatics play important roles in improving cardiac function [137], heart development, and ischemic cardiac disease [138]. Smad7 is one of the major conductors of stem cell cardiogenesis. Downregulation of Smad7 is associated with enhanced cardiogenesis in mouse embryonic and human mesenchymal stem cells [139]. Whether the stem and progenitor cells in PVAT may contribute to cardiogenesis by releasing exosomes that contain factors targeting Smad7 is unclear. Some studies have shown that EVs released from inflammatory adipocytes promote leukocyte attachment to vascular endothelial cells, suggesting the role of EVs in the setting of atherosclerosis [140]. Barberio et al. have found that adipocyte EVs regulate macrophage cholesterol homeostasis by increasing macrophage cholesterol efflux, thereby promoting the development of atherosclerosis [141]. Wang and colleagues have revealed that adipocyte EVs can promote atherosclerosis plaque vulnerability and atherosclerosis by inducing angiogenesis of the vasa vasorum in diabetic atheroprone mice [142]. For PVAT specifically, Mangiferin, a traditional Chinese medicine for diabetes, has been found to stimulate EV release from PVAT, which can improve endothelial cell migration, reduce apoptosis, and prevent inflammation by decreasing IL-6 and TNF-⍺ and blocking NF-κβ [143]. Liu and colleagues have found that PVAT-EVs could regulate the expression of macrophage cholesterol transporters [144]. Furthermore, they have demonstrated that PVAT-EVs can reduce macrophage foam cell formation and that the underlying mechanism is largely ascribed to the EV-carried miR-382–5p [145]. Nevertheless, Xie and colleagues have found that EVs isolated from visceral adipose tissues elicited the opposite effect. These EVs increase macrophage foam cell formation [146]. All of these findings demonstrate the potential roles of PVAT-EVs in vascular health and disease progression (Figure 2).

## 7. Conclusions and Perspectives

Increasing evidence indicates that PVAT plays an important role in the cardiovascular system. It is a master endocrine organ, much more than a mechanistic support to the vasculature. PVAT releases vaso-relaxing factors and anti-contractile factors to regulate the functions of endothelial and smooth muscle cells in the vasculature. It is worthwhile to note that recent studies have provided hints that PVAT can release EVs, including exosomes and microvesicles, which may either enter circulation or transfer to nearby target cells for intercellular communications. These findings open a window and shed light on the therapeutic potential of targeting PVAT and PVAT-EVs in obesity and diabetic-associated vascular diseases. However, research into PVAT-EVs is still in its infancy. There are many gaps. One of these gaps is the scarcity of studies that comprehensively explain the mechanisms and pathways through which PVAT-EVs influence the progression of different vascular diseases. On the other hand, although increasing studies show that adipose tissue-derived EV profile changes in obesity, diabetes, and vascular disorders, there is limited information on how EV biogenesis pathways are implicated in obesity and diabetic conditions. Meanwhile, how these are altered to influence the release of EVs and their cargo in individuals with obesity or diabetes is also largely unknown. Therefore, a better understanding of how EVs, including PVAT-EV biogenesis pathways, are dysregulated in obesity and diabetic conditions may allow the generation of targeted therapies to perturb EV signaling and develop new EV diagnostic tests. 

Given that PVAT becomes dysfunctional in obesity and diabetic conditions, restoring PVAT function could improve cardiometabolic risk, leading to a decreased onset of vascular damage and atherosclerosis development. Physical activity has been shown to restore PVAT function in diabetic animal models. Nevertheless, the question of whether exercise can improve the level and contents of PVAT-EVs and their potential effects on vasculature has not been studied. 

Finally, there is a shortage of clinical studies associated with PVAT and PVAT-EVs, as most existing studies utilize mice models. Clinical studies would allow a deeper exploration and understanding of the role of PVAT-EVs in cardiovascular disease. This would enable further research to utilize PVAT-EVs as biomarkers or as therapy for vascular disease, which is a potential future direction of PVAT-EV in vascular disease.

## Figures and Tables

**Figure 1 cells-13-01309-f001:**
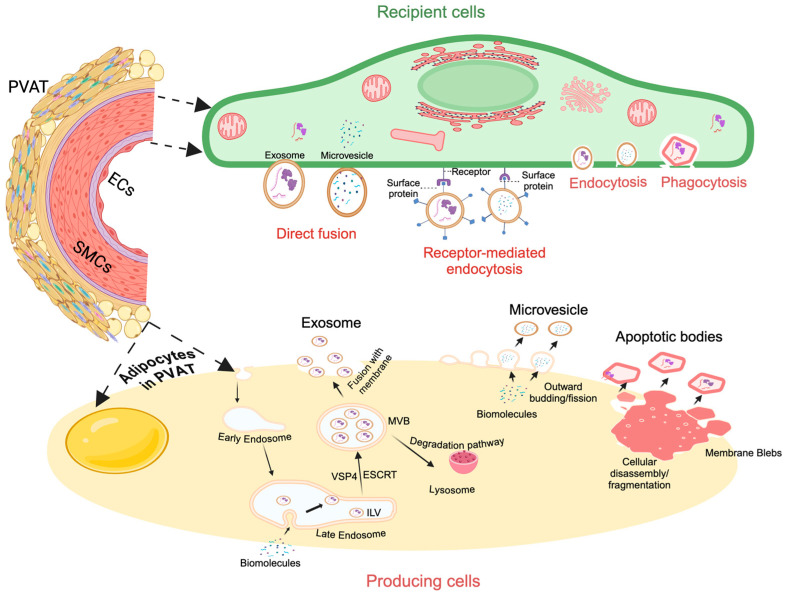
The biogenesis of the three types of PVAT-EVs and uptake pathways by the vasculature cells. The biomolecules include proteins, RNAs, DNAs, lipids, etc.

**Figure 2 cells-13-01309-f002:**
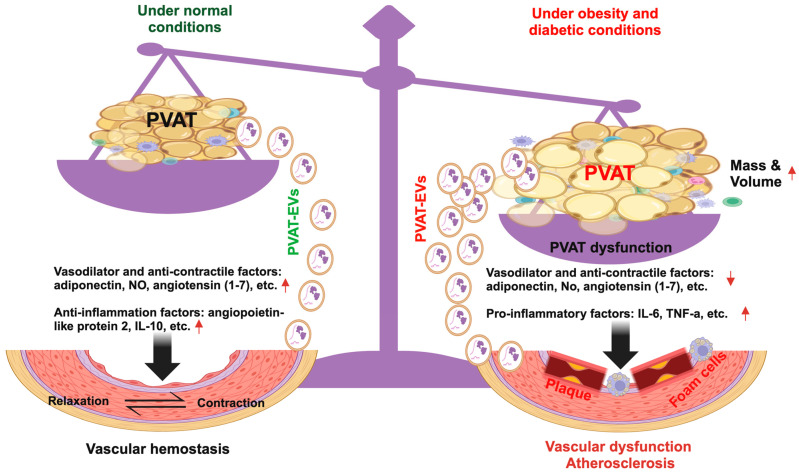
PVAT and PVAT-EVs mediated vascular function under healthy conditions and obesity/diabetic conditions.

**Table 1 cells-13-01309-t001:** Overview of anatomical and physiological changes in PVAT.

Disease	Type of Change	Effect	Reference
Obesity	Anatomical	Increase in mass	[5,6,95,96,98]
		Shift from brown to white adipocyte phenotype	[100]
		Type 1 macrophage infiltration	[101]
	Physiological	Loss of vasodilation effects	[102,104,106,108]
		Upregulated TNF-⍺, leptin, MIP1-⍺, Downregulated eNOS, adiponectin, PPAR-*γ*, FABP4,	[102]
		Decreased NO production	[105]
T2D	Anatomical	Increase in mass	[97]
		Macrophage infiltration	[107]
	Physiological	Promotion of vasoconstriction rather than vasodilation	[107]
		Downregulated adiponectin	[37,108]
		Downregulated IL-10, Upregulated INF-r, TNF-⍺, IL-6	[37]
Metabolic syndromes	Anatomical	Loss of vasodilation effects	[7,109,110]
		Shift from brown to white phenotypes	[109]
	Physiological	Increased ROS production	[7,109,110]
		Upregulated TNF-⍺, IL-1β, IL-6, IFN- *γ*, Downregulated IL-4, IL-5, IL-10, IL-13, adiponectin	[109]

**Table 2 cells-13-01309-t002:** Potential roles of adipose-derived EVs in preclinical and clinical studies.

EV Source	Disease State	Containing	Taken Up by	Effect	Reference
Adipose, general	T2D.	N/A	Brain cells	Cause synaptic damage and cognitive impairment.	[116]
	Obesity	Downregulated vascular endothelial growth factor	Vasculature	Decreased angiogenic potential	[118]
		N/A	VAT, liver, muscle	Insulin resistance, differentiation of monocytes to macrophages, production of pro-inflammatory cytokines	[10]
Human subcutaneous adipose tissue	Obesity	Proteins such as ECHA (a subunit of the trifunctional enzyme), HCDH (hydroxyacyl-coenzyme A dehydrogenase), etc.	Melanoma cells	Fatty acids oxidation	[121]
		miR-155	Macrophages	Shift toward M1 phenotype	[67]
Adipose tissue-derived macrophages	Obesity	miR-155	N/A	Modulate insulin sensitivity	[40]
PVAT	Obesity	miRNA-221	Vasculature	Stimulate vascular remodeling	[66]

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
