# Peer review of "Perivascular Adipose Tissue and Perivascular Adipose Tissue-Derived Extracellular Vesicles: New Insights in Vascular Disease"

_cells, 2024, doi:10.3390/cells13161309_

Round 1

Reviewer 1 Report

Comments and Suggestions for Authors

Sigdel et al have written a review article on the impact of perivascular fat (PVAT) and its released vesicles (PVAT-EVs) on blood vessel function in health and disease, especially in obesity, diabetes, and metabolic syndrome. This review explores PVAT and PVAT-EVs as potential theranostic targets for vascular diseases.

Major issues

1) Previous review articles have discussed the role of adipose tissue exosomes in the regulation of metabolic disorders (Curr Med Sci. 2024 Jun;44(3):463-474; Front Endocrinol (Lausanne). 2022 May 4;13:873865;Theranostics. 2020 Jun 12;10(16):7422-7435.). The authors should describe the differences between adipose tissue exosomes and those derived from perivascular adipose tissue cells.

2) The authors should give greater emphasis to the concept of selective exosome uptake by different cell types. Prior researches have convincingly demonstrated that the source of exosomes significantly impacts their uptake by target cardiac cells (Biomed Pharmacother. 2023 Nov:167:115551). These studies showed that exosomes derived from cardiac progenitor cells were preferentially taken up by cardiomyocytes and endothelial cells compared to fibroblasts and bone marrow stem cells. This issue is relevant as PVAT contains adipocyte progenitors (Physiol Rep. 2016 Oct;4(19):e12993.) . Additionally, another study (J Transl Med. 2023 May 9;21(1):313.) found that cardioprotective exosomes released from fibroblasts pre-treated with sulforaphane (an edible class I HDAC inhibitor) exhibited selective uptake by cardiomyocytes compared to exosomes from untreated cells. This additional issue is relevant as PVAT, which surrounds most blood vessels (including coronary arteries) is also rich in fibroblasts. The acetylome of fibroblasts and progenitor cells can be influenced by epigenetic food compounds (i.e.: sulforaphane), if regularly consumed, may modulate exosome biogenesis and release from PVAT fibroblasts. All abovementioned issues should be discussed in an additional paragraph.

3) Exosomes exert also cardioprotection in patients. Please discuss this relevant aspect in additional paragraph.

4) The release of exosomes is known to be dependent on cellular viability and density (please see Bioeng. Transl. Med. 2017;2:170–179). Interestingly, previous study demonstrated that selective and reversible non-thienopyridine P2Y12 inhibitor, commonly used for peripheral arterial disease (Expert Opin Pharmacother. 2020 Sep;21(13):1603-1616.) can enhance the release of adaptive exosomes from cardiac cells (Sci Rep. 2022 Apr 5;12(1):5651.) and stimulate the proliferation of vascular cells(Am J Transl Res. 2021 Dec 15;13(12):13462-13470). It would be beneficial for the authors to discuss this aspect in the context of their own observations, considering the potential impact of cellular proliferation on exosome release induced by conventional cardiovascular medication.

5) Previous study showed that exosomes released from adipose-derived stem cells can downregulate Smad7, a protein that inhibits a signaling pathway important for lymphatic vessel growth (lymphangiogenesis) (Cell Physiol Biochem. 2018;49(1):160-171) . Increased lymphangiogenesis is associated with improved cardiac function (Heart Fail Rev. 2022 Sep;27(5):1837-1856) and development (Circ Res. 2023 Apr 28;132(9):1246-1253.). Indeed, Smad7 downregulation promotes cardiogenesis as previously demonstrated in another study (PLoS One. 2010 Nov 30;5(11):e15151).Therefore, stem and progenitor cells located in the perivascular adipose tissue may contribute to cardiogenesis by releasing exosomes that contain factors targeting Smad7. The authors should discuss these relevant issues on the modulating role of PVAT-derived exosomes in the relationship between lymphangiogenesis and cardiogenesis.

6) The authors should add table summarizing relevant pre-clinical and clinical studies.

Author Response

Sigdel et al have written a review article on the impact of perivascular fat (PVAT) and its released vesicles (PVAT-EVs) on blood vessel function in health and disease, especially in obesity, diabetes, and metabolic syndrome. This review explores PVAT and PVAT-EVs as potential theranostic targets for vascular diseases.

Major issues

1) Previous review articles have discussed the role of adipose tissue exosomes in the regulation of metabolic disorders (Curr Med Sci. 2024 Jun;44(3):463-474; Front Endocrinol (Lausanne). 2022 May 4;13:873865;Theranostics. 2020 Jun 12;10(16):7422-7435.). The authors should describe the differences between adipose tissue exosomes and those derived from perivascular adipose tissue cells.

Response: We appreciate the Reviewer for this point. The breakdown of the potential functions of adipose tissue extracellular vesicles and perivascular adipose tissue extracellular vesicles are included in the revision.

2) The authors should give greater emphasis to the concept of selective exosome uptake by different cell types. Prior researches have convincingly demonstrated that the source of exosomes significantly impacts their uptake by target cardiac cells (Biomed Pharmacother. 2023 Nov:167:115551). These studies showed that exosomes derived from cardiac progenitor cells were preferentially taken up by cardiomyocytes and endothelial cells compared to fibroblasts and bone marrow stem cells. This issue is relevant as PVAT contains adipocyte progenitors (Physiol Rep. 2016 Oct;4(19):e12993.) . Additionally, another study (J Transl Med. 2023 May 9;21(1):313.) found that cardioprotective exosomes released from fibroblasts pre-treated with sulforaphane (an edible class I HDAC inhibitor) exhibited selective uptake by cardiomyocytes compared to exosomes from untreated cells. This additional issue is relevant as PVAT, which surrounds most blood vessels (including coronary arteries) is also rich in fibroblasts. The acetylome of fibroblasts and progenitor cells can be influenced by epigenetic food compounds (i.e.: sulforaphane), if regularly consumed, may modulate exosome biogenesis and release from PVAT fibroblasts. All abovementioned issues should be discussed in an additional paragraph.

Response: We thank the Reviewer for this point. We have added more relative studies regarding EV uptake (pages 9-10).

3) Exosomes exert also cardioprotection in patients. Please discuss this relevant aspect in additional paragraph.

Response: We have added more relative studies regarding EV cardioprotective effects (page 26).

4) The release of exosomes is known to be dependent on cellular viability and density (please see Bioeng. Transl. Med. 2017;2:170–179). Interestingly, previous study demonstrated that selective and reversible non-thienopyridine P2Y12 inhibitor, commonly used for peripheral arterial disease (Expert Opin Pharmacother. 2020 Sep;21(13):1603-1616.) can enhance the release of adaptive exosomes from cardiac cells (Sci Rep. 2022 Apr 5;12(1):5651.) and stimulate the proliferation of vascular cells(Am J Transl Res. 2021 Dec 15;13(12):13462-13470). It would be beneficial for the authors to discuss this aspect in the context of their own observations, considering the potential impact of cellular proliferation on exosome release induced by conventional cardiovascular medication.

Response: We have added the relative studies as suggested by the reviewer and a discussion in the revision (page 26).

5) Previous study showed that exosomes released from adipose-derived stem cells can downregulate Smad7, a protein that inhibits a signaling pathway important for lymphatic vessel growth (lymphangiogenesis) (Cell Physiol Biochem. 2018;49(1):160-171) . Increased lymphangiogenesis is associated with improved cardiac function (Heart Fail Rev. 2022 Sep;27(5):1837-1856) and development (Circ Res. 2023 Apr 28;132(9):1246-1253.). Indeed, Smad7 downregulation promotes cardiogenesis as previously demonstrated in another study (PLoS One. 2010 Nov 30;5(11):e15151).Therefore, stem and progenitor cells located in the perivascular adipose tissue may contribute to cardiogenesis by releasing exosomes that contain factors targeting Smad7. The authors should discuss these relevant issues on the modulating role of PVAT-derived exosomes in the relationship between lymphangiogenesis and cardiogenesis.

Response: We appreciate the Reviewer for this point. We have added more relative studies regarding the possible roles of PVAT-EVs in lymphangiogenesis and cardiogenesis (page 27).

6) The authors should add a table summarizing relevant pre-clinical and clinical studies.

Response: We have added two tables to summarize the reviewed articles (pages 19-20 & 23-24).

Reviewer 2 Report

Comments and Suggestions for Authors

    The description of PVAT and PVAT EVs in this article is equivalent, while PVAT is clearly more important (or comes first) since these EVs are derived from PVAT. Therefore, it is recommended to adjust the description order and move the introduction of PVAT before EVs. 

 The figure and caption describe PVAT in Figure 1 but the whole content in part of “Extracellular Vesicles (EVs)” from line 52 to 184 did not state PVAT at all. Would it be better to describe (review) the characteristics of EVs in PVAT, or at least to mention that EVs in PVAT obtain the same features in Biogenesis and Uptake Pathways as EVs in other tissue and cells in text of this part?

 The tilt direction of the balance in Figure 2 is confusing. On the right side, there is an increase in adipose tissue mass and volume, but the balance is tilted upwards. It seems like it should tilt downwards or it might be better just make it balance. Does the author believe that the proportion of fat cells is relatively light? But we believe that obesity leads to weight gain in people. I just feel that the tilt of the balance on the right is not in line with common sense.

 It takes some basketball background to understand the “the sixth man of the cardiovascular system” in first sentence of Conclusion. Could it be stated more directly?

Comments on the Quality of English Language

Good!

Author Response

1. The description of PVAT and PVAT EVs in this article is equivalent, while PVAT is clearly more important (or comes first) since these EVs are derived from PVAT. Therefore, it is recommended to adjust the description order and move the introduction of PVAT before EVs. 

Response: We thank the Reviewer for this point. We have adjusted the order of the PVAT and EVs sections in the revision.

2. The figure and caption describe PVAT in Figure 1 but the whole content in part of “Extracellular Vesicles (EVs)” from line 52 to 184 did not state PVAT at all. Would it be better to describe (review) the characteristics of EVs in PVAT, or at least to mention that EVs in PVAT obtain the same features in Biogenesis and Uptake Pathways as EVs in other tissue and cells in text of this part?

Response: We appreciate the Reviewer for this great point. We have added a statement regarding the similar features of PVAT-EVs and EVs (page 7).

3. The tilt direction of the balance in Figure 2 is confusing. On the right side, there is an increase in adipose tissue mass and volume, but the balance is tilted upwards. It seems like it should tilt downwards or it might be better just make it balance. Does the author believe that the proportion of fat cells is relatively light? But we believe that obesity leads to weight gain in people. I just feel that the tilt of the balance on the right is not in line with common sense.

Response: Sorry for the confusion and error of the scale. We corrected it (page 28).

 4. It takes some basketball background to understand the “the sixth man of the cardiovascular system” in first sentence of the Conclusion. Could it be stated more directly?

Response: We have rephrased it (page 29).

Round 2

Reviewer 1 Report

Comments and Suggestions for Authors

I have not furthe questions. However, I suggest to avoid the use of "etc" in the main text.